

# A retrospective study on expression and clinical significance of PHH3, Ki67 and P53 in bladder exophytic papillary urothelial neoplasms

Gaoxiu Qi[1,*], Jinmeng Liu[2,*], Shuqi Tao[3], Wenyuan Fan[3], Haoning Zheng[3], Meihong Wang[4], Hanchao Yang[5], Yongting Liu[6], Huancai Liu[7] and Fenghua Zhou[3]

[1] Clinical Medical College, Weifang Medical University, Weifang, Shandong, China
[2] Laboratory of Biochemistry and Molecular Biology, Weifang Medical University, Weifang, Shandong, China
[3] Department of Clinical Pathology, Weifang Medical University, Weifang, Shandong, China
[4] Department of Pathology, Hospital of PLA 80th group army, Weifang, Shandong, China
[5] Department of Pathology, Affiliated Hospital of Weifang Medical University, Weifang, Shandong, China
[6] Department of forensic medicine, Qingzhou City Public Security Bureau Interpol Brigade, Weifang, Shandong, China
[7] Affiliated Hospital of Weifang Medical University, Department of joint surgery, Weifang, Shandong, China
[*] These authors contributed equally to this work.

Corresponding authors
Huancai Liu, liuhuancaiwf@163.com
Fenghua Zhou, zhoufh@wfmc.edu.cn

## ABSTRACT

**Background.** Exophytic papillary urothelial neoplasms (EPUN) are difficult to diagnose pathologically and are well-known for their heterogeneous prognoses. Thus, searching for an objective and accurate diagnostic marker is of great clinical value in improving the outcomes of EPUN patients. PHH3 was reported to be expressed explicitly in the mitotic phase of the cell cycle, and recent studies have shown that PHH3 expression was associated with the differential diagnosis and prognosis of many tumors. However, its significance in EPUN remains unclear. This study aimed to determine the expression of PHH3 in different EPUN, compare its expression with cell-cycle related proteins Ki67 and P53, and analyze its significance in the differential diagnosis and prognostic value for high-grade papillary urothelial carcinoma (HGPUC), low-grade papillary urothelial carcinoma (LGPUC), papillary urothelial neoplasm of low malignant potential (PUNLMP) and urothelial papilloma (UP).

**Methods.** We retrospectively analyzed the pathological diagnosis and clinical features of 26 HGPUC cases, 43 LGPUC cases, 21 PUNLMP cases and 11 UP cases. PHH3, Ki67 and P53 were detected by immunohistochemistry in 101 EPUN cases samples. The cut-off values of PHH3 mitosis count (PHMC), HE mitosis count (HEMC), Ki67 and P53 in the different EPUN were determined using the ROC curve. The distribution of counts in each group and its relationship with clinical parameters and prognosis of EPUN patients were also analyzed.

**Results.** The determination coefficient ($R^2 = 0.9980$) of PHMC were more potent than those of HEMC ($R^2 = 0.9734$) in the EPUN mitotic counts microscopically by both pathologists. Of the 101 EPUN cases investigated, significant positive linear correlations were found between PHMC and HEMC, PHMC and Ki67, and HEMC and Ki67 ($P < 0.0001$). In HGPUC, LGPUC, PUNLMP and UP, a decreasing trend was observed in the median and range of PHMC/10HPFs, HEMC/10HPFs, Ki67 (%)

and P53 (%). PHMC, HEMC, Ki67 and P53 were associated with different clinical parameters of EPUN. PHMC, HEMC, Ki67 and P53 were found to exhibit substantial diagnostic values among different EPUN and tumor recurrence. Based on the ROC curve, when PHMC was >48.5/10HPFs, a diagnosis of HGPUC was more likely, and when PHMC was >13.5/10HPFs, LGPUC was more likely. In addition, when PHMC was >5.5/10HPFs, the possibility of non-infiltrating LGPUC was greater. Kaplan-Meier survival curve analysis showed that the median recurrence-free survival (RFS) for cases with PHMC > 13.5/10HPFs and HEMC > 14.5/10HPFs were 52.5 and 48 months, respectively, and their respective hazard ratio was significantly higher (Log-rank $P < 0.05$).

**Conclusion**. PHH3 exhibited high specificity and sensitivity in diagnosing EPUN. Combined with HEMC, Ki67 and P53, it can assist in the differential diagnosis of EPUN and estimate its clinical progression with high predictive value to a certain extent.

# INTRODUCTION

Urothelial neoplasm (UN) is a heterogeneous disease that occurs with varying frequency at different sites along the urothelial tract, with over 90% of bladder tumors originating from the urothelial tissues (*Lenis et al., 2020*). Presently, three types of growth patterns have been described, including exophytic papillary, inverted papillary and flat, of which the most common is exophytic papillary. Exophytic papillary urothelial neoplasms (EPUN) can be divided into high-grade papillary urothelial carcinoma (HGPUC), low-grade papillary urothelial carcinoma (LGPUC), papillary urothelial neoplasms of low malignant potential (PUNLMP) and urothelial papilloma (UP). Urothelial carcinoma can be further classified as invasive, including invasive high-grade papillary urothelial carcinoma (IHGPUC) and invasive low-grade papillary urothelial carcinoma (ILGPUC), and non-invasive, including non-invasive high grade papillary urothelial carcinoma (NIHGPUC) and non-invasive low grade papillary urothelial carcinoma (NILGPUC) (*Mikhaleva et al., 2021*). The histological morphology and classification of EPUN are variable and complex, with some histological overlap between NILGPUC, PUNLMP and UP, which is challenging to identify in some cases, leading to different treatments and patients' prognoses (*Tian et al., 2016*). Regardless of the types of EPUN, these tumors have a high postoperative recurrence rate, seriously impacting patients' survival and quality of life (*Lopez-Beltran, 2008*), thus urging the need for new biomarkers for the early diagnosis, differential diagnosis, prognosis assessment, and effective treatment of EPUN patients.

Mitotic count is an essential indicator for measuring cell proliferation activity and was shown to be valuable for EPUN identification to some extent (*Kirkali et al., 2005*). Although HE mitotic count (HEMC) is commonly used in clinical assessment, its applicability might be limited due to the lengthy time of obtaining the results and clinician's subjectivity. Comparatively, Ki67 is a DNA-binding nuclear protein expressed throughout the G1, S,

G2 and M phases of the cell cycle, and similar to HEMC, result interpretation subjectivity is its major limitation in immunohistochemistry (IHC) staining in addition to possible nonspecific staining of apoptotic cells. P53 is the gene with the highest correlation with human tumors identified so far. Since Ki67 and P53 are established cell proliferation markers widely used in various cancers (*Comperat et al., 2006*; *Erill et al., 2004*), there has been continuous research to find similar or more effective biological markers. In this regard, phosphorylated histone H3 (PHH3) is a newly studied core histone of eukaryotic cells, forming chromatin's primary component in its phosphorylated state. PHH3 is phosphorylated at Ser10 or Ser28. Since its phosphorylation at Ser10 is critical for chromatin concentration (*Eberlin et al., 2008*), it is used as a marker to determine the phase of cell cycle division (M-phase). Additionally, it has been widely used for grading and prognosis prediction in various tumors, including prostate cancer, gliomas and endometrial cancer (*Brunner et al., 2012*; *Goltz et al., 2015*; *Zhu et al., 2016*). Thus, we believe that the objective evaluation of Ki67, P53 and PHH3 in EPUN might help improve the diagnosis and treatment outcomes of EPUN patients.

With the in-depth study of the cell cycle, many scholars believe that the dysregulation of the cell cycle is closely related to the occurrence and development of various tumors because it plays an essential role in the malignant biological behavior of tumors and patients' prognosis. In addition, PHH3, Ki67 and P53 have been related to the cell cycle. Further, although many studies have shown that P53 and Ki67 are closely related to urothelial tumors, the expression of PHH3 in urothelial tumors has rarely been studied. Thus, this study aimed to detect the expressions of PHH3, Ki67 and P53 and evaluate their significance in the diagnosis and prognosis estimation of different types of EPUN.

## MATERIAL AND METHODS

### Data collection

The specimens of 101 patients were retrospectively collected from January 2015 to December 2017, comprising 26 cases of HGPUC, 43 cases of LGPUC, 21 cases of PUNLMP, and 11 cases of UP. Due to the heterogeneity of the disease, lesions involving two or more grades were graded according to the highest grade determined in the specimens. All patients were followed for 2–60 months. The follow-up time of patients without recurrence was 60 months, and the recurrence-free survival (RFS) of patients without recurrence was 60 months. Patients with recurrence were followed until the first recurrence. The study inclusion criteria were: 1. diagnosis was confirmed by pathological examination; 2. treatment naïve patients with first-time diagnosis; 3. the first treatment was transurethral resection of bladder tumor (TURBT) rather than radical cystectomy. Cases were excluded if: 1. they had missing clinical data and lost to follow-up; 2. underwent chemotherapy or radiotherapy before enrollment; and 3. had liver and kidney injury and other tumor disease history. We received a waiver of the need for informed consent from participants of our study. This study was approved by the Ethics Committee of the Affiliated Hospital of Weifang Medical University, China (approval number: wyfy-2022-ky-169).

## Immunohistochemistry

All IHC steps were conducted in the Dako platform. The paraffin-embedded tissues were cut (3μm), and pretreated at 60 °C for 2 h. For each sample, a section was conducted and stained by an indirect immunoperoxidase method with anti-human phosphorylated histone H3 (PHH3) (Ser10) antibody (polyclonal rabbit antibody; catalog No. ZA-0477; ZSGB-bio, China), anti-human Ki67 antibody (MIB1; monoclonal mouse antibody; catalog No. ZM-0166; ZSGB-bio, China), and mouse anti-human P53 antibody (monoclonal mouse antibody; catalog No. ZM-0488; ZSGB-BIO, Beijing, China). After washing and incubation, the sections were colored and mounted.

## Data analysis

Using light microscopy, positivity for PHH3, Ki67 and P53 was based on brown-yellow particles identified in the nucleus. Two senior pathologists, double-blinded to all indicators, performed the assessment, and the average of the two was taken for analysis. Briefly, PHMC, HEMC, P53 and Ki67 were first located in hot spots at $100\times$ and then accurately counted in 10 fields consecutively at $400\times$. For PHMC and HEMC, the total number of positive cells in 10 fields was calculated. For Ki67 and P53, the average percentage of positive cells in 10 fields was calculated. P53 $\geq$ 50% was considered mutant.

## Statistical analysis

The SPSS v23.0 software and GraphPad Prism v8.0.2 software were used for statistical analysis and chart drawing. X2 test was used to analyze the differences between PHMC, HEMC, Ki67 and P53 expression and clinical data parameters. Spearman rank correlation coefficient was used to analyze the linear correlation between PHMC, HEMC, Ki67 and P53 and the different pathologists. ROC curve and Youden index were used to analyze the diagnostic value, cut-off value, specificity and sensitivity of PHMC, HEMC, Ki67 and P53 in different urothelial tumors and patients with or without recurrence. Kaplan–Meier survival curves were drawn to analyze the relationship between recurrence risk and recurrence-free survival time, and the Log-rank test was used to test the difference. $P < 0.05$ was used as the criterion for statistical significance.

# RESULTS

## Considerations, advantages, and diagnostic criteria for PHH3 mitosis count

PHH3 was expressed in all stages of nuclear mitosis, including prophase, metaphase, anaphase, telophase, and its positivity was determined by the brown-yellow coloration of the nucleus. During prophase, the nuclei lose their nuclear membranes, and chromatin condenses into dense clumps (Fig. 1A), after which chromatin separates and redistributes (Fig. 1B) into dipolar, tripolar or multipolar divisions (Figs. 1C–1E). During the anaphase of mitosis, chromatin separates, and two identical sets of chromatin appear at the poles of the cell (Fig. 1F). In telophase, nucleus divides into two and the daughter cells form after cytokinesis (Fig. 1G). Compared with the mitotic count observed under the HE, PHMC can be more intuitively and conveniently.

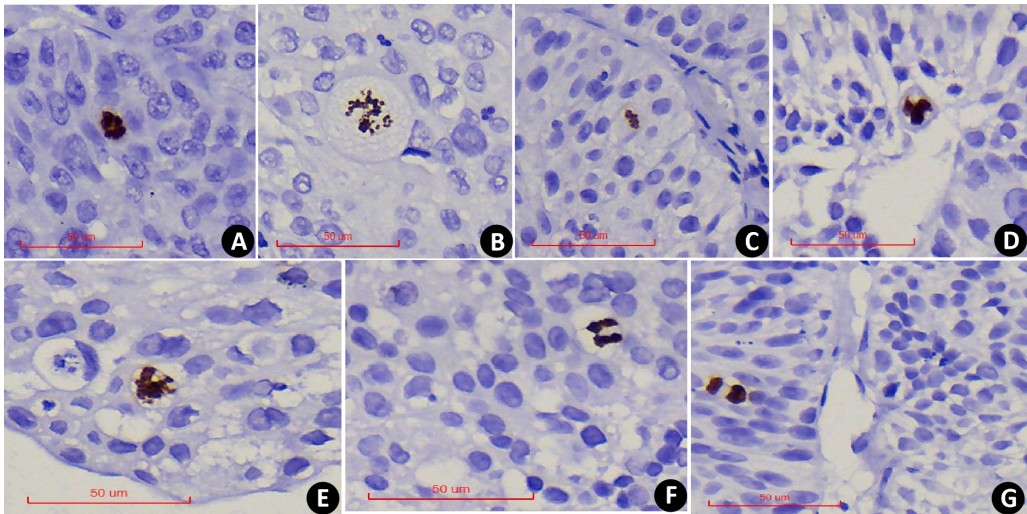

**Figure 1** **PHH3x400: Examples of PHH3 positive mitotic figures: nuclear envelope deletion and chromosome condensation.** (A) Prophase. (B) Prophase to metaphase, chromatin separates and redistributes. (C) Dipolar mitosis. (D) Tripolar mitosis. (E) Multipolar mitosis. (F) Anaphase of mitosis. (G) Telophase of mitosis.

In comparison, serial section assessment showed that PHH3 excluded nuclear fragmentation (Figs. 2A–2B) and accounted for the extrusion artifact of the nuclear fission image (Fig. 2C). In addition, PHH3 was also expressed in the G2 phase of the cell cycle (Fig. 2D) and exhibited nonspecific staining (Fig. 2E). Positive cells had delicate chromatin and intact nuclear envelope, which were excluded from counting. In addition, PHH3 was found to be expressed in interstitial cells or lymphocytes in EPUN (Fig. 2F) but not in cells of epithelial origin and were excluded.

PHMC and HEMC were both diagnosed and counted by two senior pathologists. The results showed a significant positive correlation between the mitotic counts of PHH3 and HE microscopically by both pathologists ($P<0.0001$). Additionally, the Spearman correlation coefficient ($r_s = 0.9990$) and determination coefficient ($R^2 = 0.9980$) of PHMC were more potent than those of HEMC ($r_s = 0.9866$) ($R^2 = 0.9734$) (Fig. 3). Further, we also found that the time to count PHMC was significantly shorter than HEMC by both pathologists. In HGPUC, LGPUC, PUNLMP and UP, time consumed of the PHMC were 1.4, 1.3, 1.1 and 1.1 times higher than HEMC.

### Linear correlation between PHMC, HEMC, Ki67 and P53

Of the 101 EPUN cases investigated, significant positive linear correlations were found between PHMC and HEMC, PHMC and Ki67, and HEMC and Ki67 ($P < 0.0001$). Additionally, their respective Spearman correlation coefficients were 0.9960, 0.9509 and 0.9469, and their determination coefficient $R^2$ were 0.9920, 0.9042 and 0.8966 (Fig. 3), respectively. Among them, the positive linear correlation and model fit between PHMC and HEMC was the strongest, followed by PHMC and Ki67, while those of HEMC and

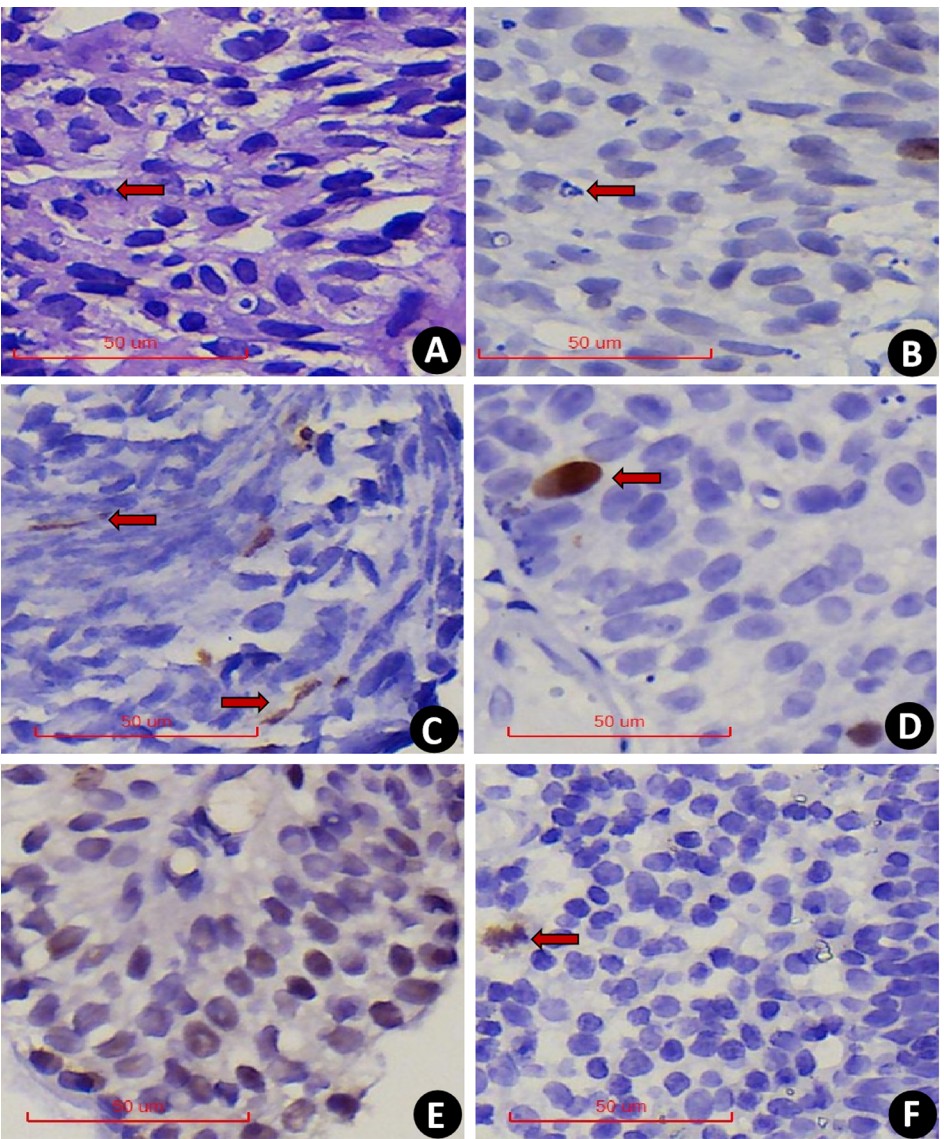

**Figure 2** **(A–F) 400x PHH3 excludes the diagnosis.** (A) Nuclear fragmentation was seen on HE (The red arrow points to). (B) Serial sections were negative for PHH3(The red arrow points to). (C) PHH3 shows extrusion mitotic figure. (D) The G2 phase were excluded. (E) Nonspecific pigmentation was excluded. (F) Positive interstitial cells were excluded.

Ki67 were the weakest. However, we found no linear correlation between PHMC and P53, HEMC and P53 or Ki67 and P53 ($P > 0.05$).

## Expression and distribution of PHMC, HEMC, Ki67 and P53 in different EPUN

PHMC, HEMC, Ki67 and P53 positive cells were mainly distributed the basement membrane of PUNLMP, UP and normal urothelium (NU) epithelium (Fig. 4). In the 26 HGPUC cases, 43 LGPUC cases, 21 PUNLMP cases and 11 UP cases, the median

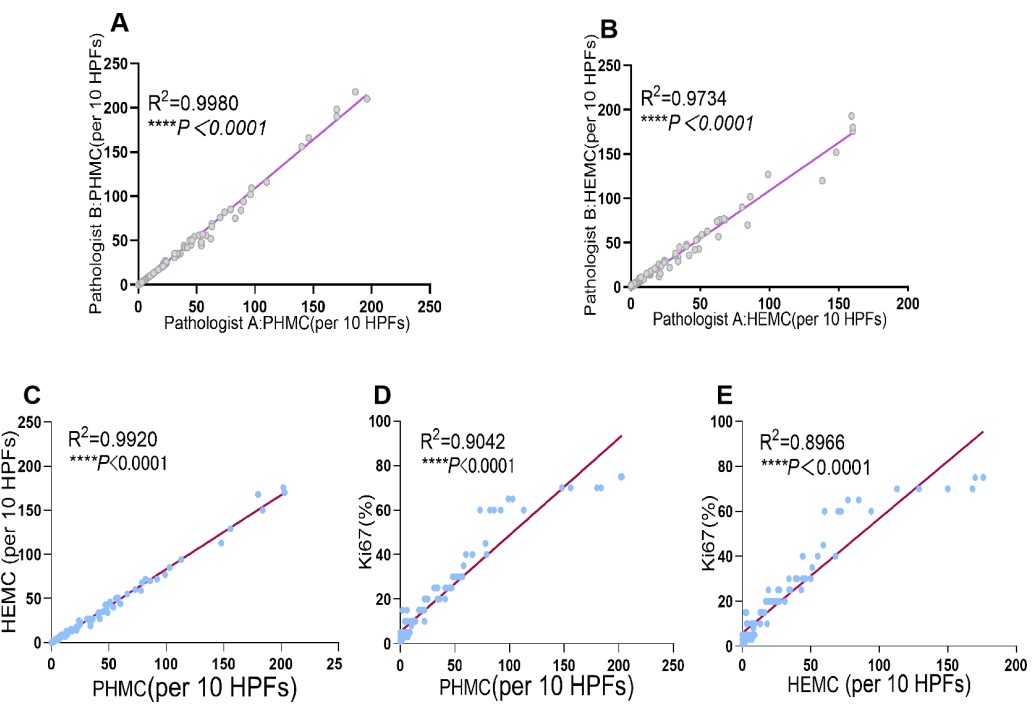

**Figure 3** **Linear regression curves of PHMC and HEMC by pathologist A and pathologist B. Linear regression curves between PHMC and HEMC, PHMC and Ki67, HEMC and Ki67.** (A) Linear regression curve of PHMC by pathologist A and pathologist B. (B) Linear regression curve of HEMC by pathologist A and pathologist B. (C) Linear regression curve between PHMC and HEMC. (D) Linear regression curve between PHMC and Ki67. (E) Linear regression curve between HEMC and Ki67. HEMC: HE mitotic count. PHMC: PHH3 mitotic count. HPFs: high power fields. ****$P < 0.0001$.

(range) of PHMC, HEMC, Ki67 and P53 are shown in Table 1. The median and range of PHMC in HGPUC, LGPUC, PUNLMP and UP showed a decreasing trend, similar to HEMC and Ki67, while no significant trend was observed for P53. We also found that the median and range of HEMC were lower than those of PHMC.

In the vast majority of HGPUC, PHMC was >30/10HPFs. The PHMC of most LGPUC cases was >15/10HPFs. For all PUNLMP and UP, PHMC was <15/10HPFs (Fig. 5A). HEMC and PHMC distributions were generally consistent (Fig. 5B). In most HGPUC, Ki67 was >20%. In most LGPUC and PUNLM, Ki67 was >10%. In all UP cases, the percentage of Ki67 was <10% (Fig. 5C). Further, the percentage of P53 in 46% of HGPUC cases was >50%. In the vast majority of LGPUC, PUNLMP and UP cases, the percentage of P53 was <50% (Fig. 5D). The distribution of PHMC, HEMC, Ki67 and P53 in HGPUC, LGPUC, PUNLMP and UP showed a decreasing trend.

There were significant differences in PHMC between HGPUC and LGPUC, HGPUC and PUNLMP, HGPUC and UP, LGPUC and PUNLMP, and LGPUC and UP ($P < 0.05$) (Fig. 5A). The differences in HEMC, Ki67 and P53 among different tumors are shown in Fig. 6.

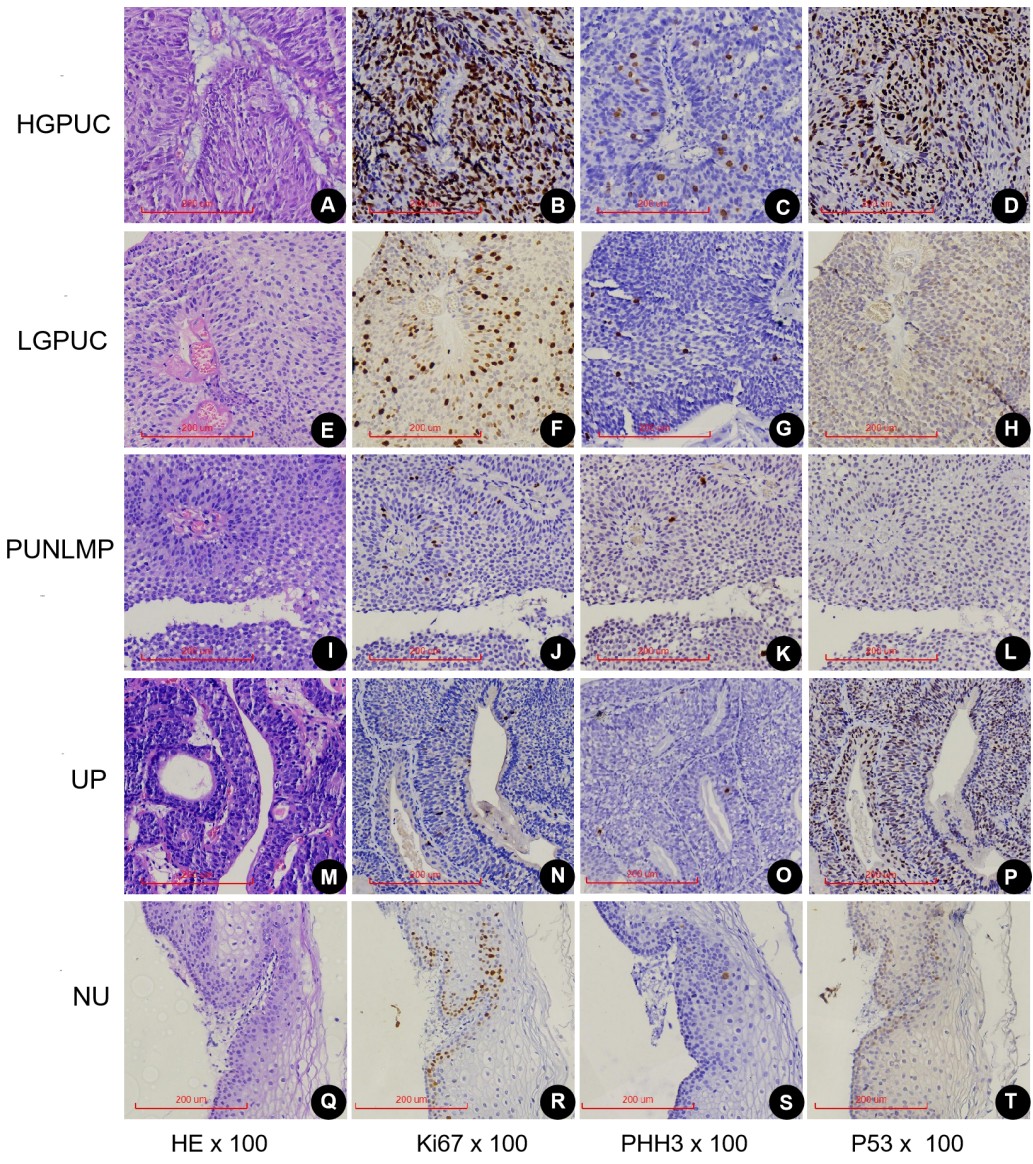

**Figure 4 Expression of Ki67, PHH3 and P53 in different exophytic papillary urothelial neoplasms and normal urothelial tissues.** Including HE(A), Ki67(B), PHH3(C) and P53(D) of the same HGPUC sample. Including HE(E), Ki67(F), PHH3(G) and P53(H) of the same LGPUC sample. Including HE(I), Ki67(J), PHH3(K) and P53(L) of the same PUNLMP sample. Including HE(M), Ki67(N), PHH3(O) and P53(P) of the same UP sample. Including HE(Q), Ki67(R), PHH3(S) and P53(T) of the same NU sample. HGPUC, high grade papillary urothelial carcinoma . LGPUC, low grade papillary urothelial carcinoma . PUNLMP, papillary urothelial neoplasms of low premalignant potential. UP, urothelial papilloma. NU, normal urothelium.

## Diagnostic significance and the cut-off value of PHMC, HEMC, Ki67 and P53 in different EPUN and recurrence

PHMC had substantial diagnostic values for differentiating between HGPUC and LGPUC (Fig. 6A), LGPUC and PUNLMP (Fig. 6B), LGPUC and UP (Fig. 6C), NILGPUC and PUNLMP (Fig. 6D), NILGPUC and UP (Fig. 6E), and estimating the risk of recurrence

**Table 1** Comparison of the diagnostic value including cut-off values and corresponding sensitivity, specificity, and Youden index of PHMC, HEMC, and Ki67 in different exophytic papillary urothelial neoplasms and recurrence.

| EPUN 1 | HGPUC | LGPUC | LGPUC | NILGPUC | NILGPUC | Recurrence |
|---|---|---|---|---|---|---|
| EPUN 2 | LGPUC | PUNLMP | UP | PUNLMP | UP | No Recurrence |
| **PHMC** | | | | | | |
| cut-off | 48.5 | 13.5 | 3 | 5.5 | 3 | 13.5 |
| Sensitivity | 0.731 | 0.558 | 0.791 | 0.75 | 0.781 | 0.806 |
| Specificity | 0.884 | 1 | 0.818 | 0.762 | 0.818 | 0.671 |
| Youden index | 0.651 | 0.558 | 0.609 | 0.512 | 0.599 | 0.477 |
| **HEMC** | | | | | | |
| cut-off | 29 | 13.5 | 2.5 | 9.5 | 2.5 | 14.5 |
| Sensitivity | 0.769 | 0.512 | 0.791 | 0.5 | 0.781 | 0.774 |
| Specificity | 0.791 | 1 | 0.818 | 0.923 | 0.818 | 0.7 |
| Youden index | 0.56 | 0.512 | 0.609 | 0.452 | 0.599 | 0.474 |
| **Ki67** | | | | | | |
| cut-off | 27.5 | 4 | 2.5 | 4 | 2.5 | 9 |
| Sensitivity | 0.731 | 0.86 | 1 | 0.844 | 1 | 0.903 |
| Specificity | 0.86 | 0.857 | 0.727 | 0.857 | 0.727 | 0.571 |
| Youden index | 0.591 | 0.717 | 0.727 | 0.701 | 0.727 | 0.474 |
| **P53** | | | | | | |
| cut-off | 67.5 | | | | | |
| Sensitivity | 0.462 | | | | | |
| Specificity | 0.93 | | | | | |
| Youden index | 0.392 | | | | | |

**Notes.**

EPUN, exophytic papillary urothelial neoplasms.

(Fig. 6F). The area under the ROC curve was relatively large, and the difference was statistically significant ($P < 0.05$). Similar findings were observed for HEMC and Ki67. Comparatively, P53 had a weak diagnostic value only in HGPUC and LGPUC, and the area under the ROC curve was relatively small ($P < 0.05$) (Fig. 6A).

The cut-off value of PHH3 for diagnosing HGPUC and LGPUC, LGPUC and PUNLMP, and LGPUC and UP was 48.5/10 HPFs, 13.5/10 HPFs and 3/10 HPFs, respectively. The cut-off values of PHH3 for NILGPUC and PUNLMP, NILGPUC and UP were 5.5/10 HPFs and 3/10 HPFs. The cut-off values of PHMC, HEMC and Ki67 for differentiating between non-recurrence and recurrence were 13.5/10 HPFs, 14.5/10 HPFs and 9%, respectively. PHMC had a higher value in distinguishing HGPUC from LGPUC (Youden index, 0.651), LGPUC and PUNLMP (Youden index, 0.558), LGPUC and UP (Youden index, 0.609), NILGPUC and PUNLMP (Youden index, 0.512), NILGPUC and UP (Youden index, 0.599), and recurrence (Youden index, 0.477). Detailed Cut-off values and corresponding specificity, sensitivity and Youden index are shown in Table 2.

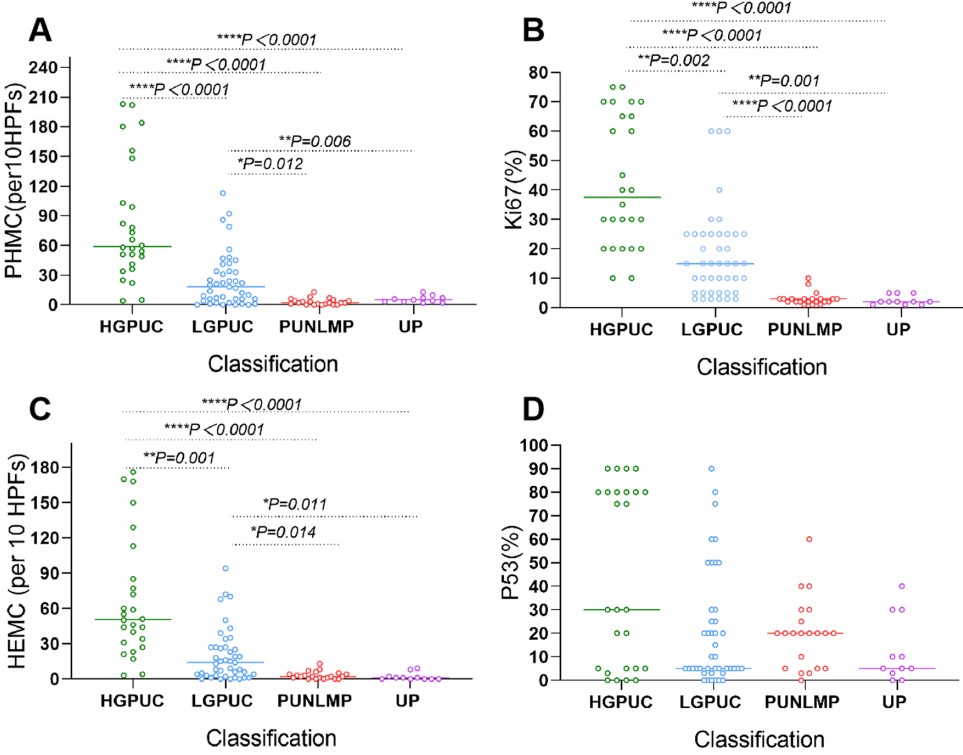

**Figure 5** **Scatter plot shows the distribution of PHMC, HEMC, Ki67 and P53 in different exophytic papillary urothelial neoplasms. Lines represent medians.** (A) Scatter plot shows the distribution of PHMC in different exophytic papillary urothelial neoplasm. (B) Scatter plot shows the distribution of HEMC in different exophytic papillary urothelial neoplasm. (C) Scatter plot shows the distribution of Ki67 in different exophytic papillary urothelial neoplasm. (D) Scatter plot shows the distribution of P53 in different exophytic papillary urothelial neoplasm. HEMC: HE mitotic count. PHMC: PHH3 mitotic count. per 10 HPFs: per 10 high power fields. HGPUC, high grade papillary urothelial carcinoma. LGPUC, low grade papillary urothelial carcinoma. PUNLMP, papillary urothelial neoplasms of low premalignant potential. UP, urothelial papilloma. *$P < 0.05$, **$P < 0.01$, ***$P < 0.001$, ****$P < 0.0001$.

**Table 2** **The median and range of PHMC, HEMC, Ki67, and P53 in the 26 HGPUC cases, 43 LGPUC cases, 21 PUNLMP cases and 11 UP cases.**

| Variates | n | PHMC (per10HPFs) | HEMC (per10HPFs) | Ki67 (%) | P53 (%) |
|---|---|---|---|---|---|
| HGPUC | range | 4-203 | 3-176 | 10-75 | 0-90 |
| | median | 59 | 51 | 38 | 30 |
| LGPUC | range | 0-113 | 0-94 | 3-60 | 0-90 |
| | median | 18 | 14 | 15 | 5 |
| PUNLMP | range | 0-13 | 0-13 | 1-10 | 0-60 |
| | median | 2 | 2 | 3 | 20 |
| UP | range | 0-9 | 0-9 | 1-5 | 0-40 |
| | median | 1 | 1 | 2 | 5 |

**Notes.**

HEMC, HE mitotic count; PHMC, PHH3 mitotic count.

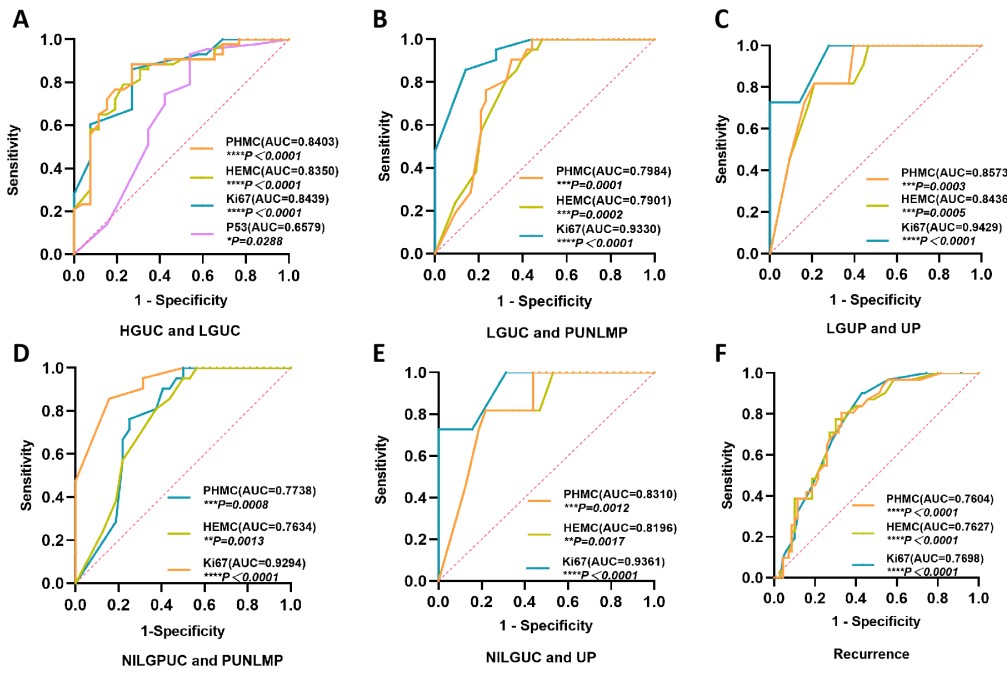

**Figure 6** **The ROC curves of PHMC, HEMC, Ki67 and P53 in the diagnosis of different exophytic papillary urothelial neoplasms and the prediction of recurrence.** (A) The ROC curve of PHMC, HEMC, Ki67 and P53 in the diagnosis of HGPUC and LGPUC. (B) The ROC curve of PHMC, HEMC, and Ki67 in the diagnosis of LGPUC and PUNLMP. (C) The ROC curve of PHMC, HEMC, and Ki67 in the diagnosis of LGPUC and UP. (D) The ROC curve of PHMC, HEMC, and Ki67 in the diagnosis of NILGPUC and PUNLMP. (E) The ROC curve of PHMC, HEMC, and Ki67 in the diagnosis of NILGPUC and UP. (F) The ROC curve of PHMC, HEMC, and Ki67 in the diagnosis of recurrence and non-recurrence. AUC: area under the curve. HGPUC, high grade papillary urothelial carcinoma. LGPUC, low grade papillary urothelial carcinoma. PUNLMP, papillary urothelial neoplasms of low premalignant potential. UP, urothelial papilloma. NILGPUC, non-invasive low grade papillary urothelial carcinoma. *$P < 0.05$, **$P < 0.01$, ***$P < 0.001$, ****$P < 0.0001$.

## Relationship between PHMC, HEMC, Ki67, P53 and clinicopathological features of EPUN

There were significant differences in PHMC, HEMC and Ki67 among the age, tumor volume, recurrence, death, depth of invasion and clinical stage of EPUN patients ($P < 0.05$). However, no significant difference was observed in gender, drinking history and smoking history ($P > 0.05$). A cut-off value of 13.5/10 HPFs, 14.5/10 HPFs, 9% and 50% for PHMC, HEMC, Ki67 and P53 was used to determine the distribution frequency of different clinicopathological features, respectively. The results showed that PHMC > 13.5/10HPFS accounted for most patients aged ≥65, had relapsed and died. With the increased tumor size, infiltration depth and clinical stage, the percentage of patients with PHMC > 13.5/10 HPFs increased gradually. The relationship between the other clinicopathological features and PHMC, HEMC, Ki67 and P53 is shown in Table 3.

In this study, 101 EPUN patients were followed up for 2 to 60 months. Of them, we found that 8 (7.9%) patients had died and 31 (30.7%) had relapsed, all of which occurred in HGPUC and LGPUC. Among all 69 patients diagnosed with HGPUC and LGPUC, the

**Table 3** Association of PHMC, HEMC, Ki67, and P53 with 101 clinical parameters of exophytic papillary urothelial neoplasms.

| Variates | | n | PHMC/10HPFs | | | HEMC/10HPFs | | | Ki67(%) | | | P53(%) | | |
|---|---|---|---|---|---|---|---|---|---|---|---|---|---|---|
| | | | <13.5 | >13.5 | *P*-Value | <14.5 | >14.5 | *P*-Value | ≤9 | >9 | *P*-Value | <50 | ≥50 | *P*-Value |
| Gender | Male | 78 | 40 | 38 | 0.813 | 42 | 36 | 0.637 | 32 | 46 | 0.634 | 60 | 18 | 0.775 |
| | Female | 23 | 13 | 10 | | 14 | 9 | | 11 | 12 | | 19 | 4 | |
| Age(years) | <65 | 44 | 30 | 14 | 0.009** | 31 | 13 | 0.009** | 25 | 19 | 0.015* | 37 | 7 | 0.234 |
| | ≥65 | 57 | 23 | 34 | | 25 | 32 | | 18 | 39 | | 42 | 15 | |
| History of drinking | N0 | 67 | 34 | 33 | 0.677 | 36 | 31 | 0.676 | 29 | 38 | 0.840 | 52 | 15 | 0.836 |
| | Yes | 34 | 19 | 15 | | 20 | 14 | | 14 | 20 | | 27 | 7 | |
| History of smoking | N0 | 61 | 35 | 26 | 0.308 | 36 | 25 | 0.417 | 31 | 30 | 0.043* | 47 | 14 | 0.808 |
| | Yes | 40 | 18 | 22 | | 20 | 20 | | 12 | 28 | | 32 | 8 | |
| Tumor size(mm²) | <4 | 80 | 47 | 33 | 0.025* | 49 | 31 | 0.023* | 40 | 40 | 0.013** | 67 | 13 | 0.028* |
| | ≥4and<15 | 13 | 5 | 8 | | 6 | 7 | | 2 | 11 | | 7 | 6 | |
| | ≥15 | 8 | 1 | 7 | | 1 | 7 | | 1 | 7 | | 5 | 3 | |
| Cases of recurrence | No | 70 | 47 | 23 | 0.000*** | 49 | 21 | 0.000*** | 40 | 30 | 0.000*** | 57 | 13 | 0.297 |
| | Yes | 31 | 6 | 25 | | 7 | 24 | | 3 | 28 | | 22 | 9 | |
| Cases of death | No | 93 | 52 | 41 | 0.029* | 55 | 38 | 0.021* | 43 | 50 | 0.019* | 74 | 19 | 0.367 |
| | Yes | 8 | 1 | 7 | | 1 | 7 | | 0 | 8 | | 5 | 3 | |
| Depth of invasion | No | 65 | 48 | 17 | 0.000*** | 51 | 14 | 0.000*** | 41 | 24 | 0.000*** | 60 | 5 | 0.000*** |
| | Lamina propria | 26 | 4 | 22 | | 4 | 22 | | 2 | 24 | | 12 | 14 | |
| | Muscular layer | 5 | 1 | 4 | | 1 | 4 | | 0 | 5 | | 3 | 2 | |
| | Serosa layer | 5 | 0 | 5 | | 0 | 5 | | 0 | 5 | | 4 | 1 | |
| Clinical stage | UP and PUNLMP | 32 | 32 | 0 | 0.000*** | 32 | 1 | 0.000*** | 31 | 2 | 0.000*** | 31 | 2 | 0.029* |
| | Ta+Tis+T1 | 59 | 20 | 39 | | 23 | 35 | | 12 | 46 | | 41 | 17 | |
| | T2+T3+T4 | 10 | 1 | 9 | | 1 | 9 | | 0 | 10 | | 7 | 3 | |

**Notes.**
*$P < 0.05$
**$P < 0.01$
***$P < 0.001$
HEMC, HE mitotic count; PHMC, PHH3 mitotic count.

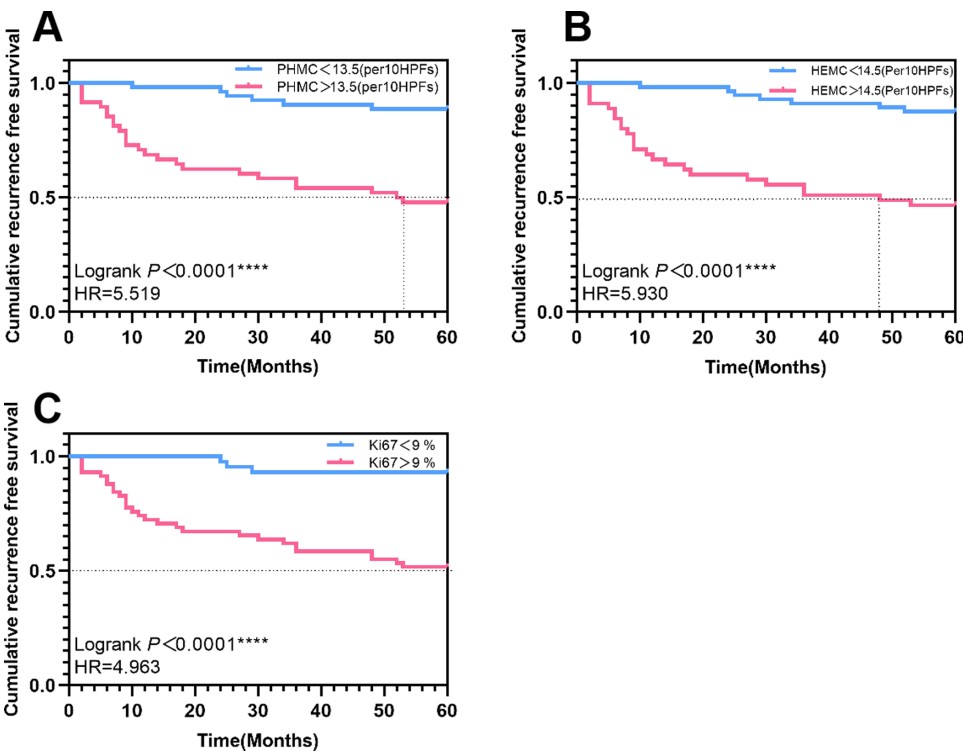

**Figure 7** **The Kaplan–Meier survival curves of patients with exophytic papillary urothelial neoplasms with different expressions of PHMC, HEMC and Ki67. Lines represent the median recurrence-free survival.** (A) The Kaplan–Meier survival curve of PHMC. (B) The Kaplan–Meier survival curve of HEMC. (C) The Kaplan–Meier survival curve of Ki67. HEMC: HE mitotic count. PHMC: PHH3 mitotic count. HR: Hazard Ratio. ****$P < 0.0001$.

recurrence rate was 31/69(44.9%). Additionally, PHMC, HEMC, Ki67, death status, depth of invasion and clinical stage were correlated with recurrence ($P < 0.05$), while P53 and other clinical parameters were not correlated with recurrence ($P > 0.05$).

A cut-off value of 13.5/10 HPFs, 14.5/10 HPFs and 9% for PHMC, HEMC and Ki67 was used as baseline for univariate survival analysis. K-M curves were plotted, and hazard ratios were calculated. During the follow-up, 48 (47.5%) cases had PHMC>13.5/10HPFs, 45(44.6%) had HEMC>14.5/10 HPFs, and 58 (57.4%) had Ki67>9%. The median recurrence-free survival (RFS) for patients with PHMC>13.5/10 HPFs and HEMC>14.5/10 HPFs were 52.5 and 48 months, respectively. The recurrence risk of patients with PHMC>13.5/10 HPFs was 5.519 times that of patients with PHMC<13.5/10 HPFs (Logrank $P < 0.0001$) (Fig. 7A). The hazard ratios and statistical significance of HEMC and Ki67 are shown in Figs. 7B–7C.

## DISCUSSION

The occurrence and development of tumors involve many genetic factors, among which many possible mechanisms often directly or indirectly affect normal cell cycle regulation. Spontaneous cell division and proliferation are the main causes of malignant tumor

occurrence. Thus, detecting tumor cell proliferation is of great significance for assisting in cancer diagnosis, evaluating patients' prognosis and guiding treatment (*Ribalta et al., 2004*). EPUN is one of the major diseases that seriously threaten human health. Some cases develop repeated relapses, while others succumb to metastasis (*Sung et al., 2021*). Thus, to improve the curative treatment of EPUN, it is necessary to achieve early diagnosis and provide timely treatment. Despite advances in understanding tumor biology, assessing EPUN recurrence risk is still largely dependent on clinical features, including tumor grade, depth of invasion and size (*Zhang & Zhang, 2015*). Since changes at the molecular level usually occur much earlier than changes at the morphological level, great emphasis has been laid on identifying molecular biomarkers capable of indicating the biological behavior of tumors, predicting the risk of recurrence and providing more accurate survival estimation, especially in EPUN.

Traditional mitotic counting under HE is still widely used, and M-phase cells can be identified by their characteristic morphology. At high magnification, typical M-phase chromatin is stained with corrugated blue-purple globules by HE, while atypical M-phase chromatin is stained with asymmetrical, tripolar or multipolar patterns (*Kim et al., 2017*). However, due to the influence of many factors, even for highly skilled pathologists, this diagnostic approach is time-consuming and subjective (*Fukushima et al., 2009*). The advantages of PHMC over HEMC are as follows: (1) regular HE usually does not count cells at the end of the division phase; (2) apoptotic cells in the wavy blue-purple globules and necrotic cells with deeply stained nuclei are similar to mitosis, while dense cells, apoptosis, distorted tumor infiltration, crush artifacts and poorly fixed specimens make it challenging to accurately count mitosis; (3) regular HE to find the highest area of subjectivity, mitosis and PHH3 can quickly help us find the highest active sites of mitosis because identifying mitotic figure need relatively high magnification. Even under low magnification, immune staining cells can more accurately determine the mitosis of the most active sites. Therefore, compared to HE, PHH3 has a more accurate and sensitive immune response during mitotic counting, requires less time for mitotic measurement, and has a lower inter-observer rate of change, which is of high practical significance (*Angi et al., 2011*; *Bosch et al., 2017*; *Tapia et al., 2006*). The following points should be noted when using the PHH3 routine count. First, most studies used the number of 10 high-power fields (*Mirzaiian et al., 2020*), and few have used the positive ratio of expression (*Shin et al., 2015*). Second, cells are gradually phosphorylated during chromatin condensation from the late G2 phase to the M-phase. The difference in staining pattern allows the observer to distinguish and exclude cells at the G2 phase (with the intact nuclear envelope and speckled nuclei staining) from cells at the M-phase (diffuse, lumpy staining). Third, lymphocyte and interstitial cells in the mitotic phase also have positive PHH3 staining, which needs to be carefully observed and excluded. Fourth, PHMC is usually higher than HEMC (*Cui et al., 2015*; *Tsuta et al., 2011*). In our investigated EPUN cases, PHMC was 1.2 times higher than HEMC. In PUNLMP and UP, there was less difference in PHMC than in HEMC, possibly because there were fewer mitotic images and sparse cells in PUNLMP and UP, which were easier to recognize under HE. The above was also related to the high consistency of hotspots found by PHMC, while the opposite was true for urothelial carcinoma, which tends to miss in HEMC.

Accurate staging and grading of EPUN are vital in determining the optimal treatment. The histopathological classification of EPUN by the WHO/ISUP system is mainly based on cell atypia and structural atypia (*Lamm, 2007*). UP has thin nipples, umbrella cells, a small number of mitotic images mainly concentrated in the vice stratum basale, and no epithelial layer thickening, polarity disorder and cell atypia. Based on UP in PUNLMP, epithelial cells were thickened, with no or mild atypia, few mitotic images and located at the vice stratum basale. However, LGPUC epithelial cells were significantly thickened and slightly disordered, with mild cell atypia, more mitotic images, an upward shift of mitotic image, a thick fusion of the papilla, and the disappearance of umbrella cells. Based on LGPUC, HGPUC showed more disordered polarity and moderate to severe atypia (*Tavora & Epstein, 2008*). EPUN progression refers to infiltration into the lamina propria, muscular layer or metastasis. UP does not usually recur or progress. PUNLMP may have a low probability of recurrence risk but no progression and no specific death, while LGPUC and HGPUC have a high risk of progression, resulting in death in some cases. Thus, accurately differentiating between PUNLMP and LGPUC could avoid labeling patients as having cancer and decrease the underlying psychological impact on the patients and their relatives. PUNLMP can be differentiated from UP because PUNLMP has a shallow risk of progression biologically but is not entirely benign (*Samaratunga, Makarov & Epstein, 2002*). However, due to the similarity of the morphology of UP, PUNLMP and NILGPUC, histological grading is subjective, and differentiation is difficult in some cases. The location and number of mitotic image distribution are essential diagnostic references. Compared with HEMC, PHH3 can provide a more sensitive and accurate mitotic index. In this study, the ROC curve and Youden index were used to determine the expressions of PHMC, HEMC, Ki67 and P53 in different EPUN. We found that when PHMC was >48.5/10HPFs, a diagnosis of HGPUC was more likely, when PHMC was >13.5/10HPFs, a diagnosis of LGPUC was more likely, and when PHMC was >5.5 /10HPFs, a diagnosis of NILGPUC was more likely. The above suggests that PHH3 could be used to differentiate the histological classification of tumors and had potential differential diagnostic significance. The proposed cut-off value might provide a referential value for pathologists to assist in the differential diagnosis. PHH3 can assist in the differential diagnosis of gastrointestinal stromal tumors, pancreatic neuroendocrine tumors, benign and malignant melanoma and other tumors (*Alkhasawneh et al., 2015*; *Nasr & El-Zammar, 2008*; *Tracht, Zhang & Peker, 2017*). These results support the potential application of PHH3 in the differential diagnosis of EPUN.

EPUN tends to be more prevalent in males, which was consistent with our study, whereby we observed a male-to-female ratio of about 3:1. Smoking could significantly increase the risk of developing EPUN and has been reported as the most critical risk factor (*Freedman et al., 2011*). Among the 26 HGPUC patients, more than 50% had a history of smoking. However, it should be noted that the risk of EPUN recurrence and tumor progression could be associated with various histopathological factors (*Mertens et al., 2022*). The expression of PHMC, HEMC and Ki67 significantly differed in patient age, tumor size, depth of invasion, clinical stage, recurrence, and patient's death status. However, there was no difference between genders, drinking history and smoking history, which might be related to the small number of samples analyzed in this study, thereby urging the need for

further investigations to validate these findings. Compared with P53, PHMC, HEMC and Ki67 were associated with more clinical parameters and were statistically correlated with recurrence, all of which were independent predictors of EPUN prognosis. It is speculated that the mitotic count and proliferation index might have more predictive value than P53 in EPUN. Numerous studies have shown that most urothelial cancers are superficial lesions with a good prognosis and more than 50% relapse within two years after surgery (*Wang et al., 2014*). In this study, 69 patients with HGPUC and LGPUC were followed up for 2–60 months, and about 50% had relapsed. We also found that PHMC, HEMC and Ki67 were highly correlated, suggesting they might have good repeatability in assessing the prognosis of EPUN patients. So far, many studies have shown that PHH3 is expressed in many malignancies, including breast and ovarian cancer (*Aune et al., 2011*; *Bossard et al., 2006*). Previous literature has demonstrated that patients with high PHH3 high expression had lower survival rates, higher recurrence rate and higher mortality, which were consistent with our follow-up results in this group of patients.

## CONCLUSIONS

In summary, our results suggest PHH3 as a promising specific marker for mitotic count that can not only effectively eliminate the interference caused by subjective and objective factors but also be time-saving, intuitive and accurate in the determination of mitotic count and improve the repeatability and consistency of diagnosis. PHH3 combined with HEMC, Ki67 and P53 can assist in the differential diagnosis of EPUN. Its high expression was more likely to be seen in EPUN with high invasive and recurrence rates, which has a high practical value in the judgment of tumor recurrence and progression. However, a larger sample size is needed to develop a more accurate quantitative method.

### Funding

This work was supported by the Shandong Province National Natural Science Foundation (ZR2020MH149) and Scientific research project of Weifang Health Commission (Grant No. WFWSJK-2022-231). The funders had no role in study design, data collection and analysis, decision to publish, or preparation of the manuscript.

### Grant Disclosures

The following grant information was disclosed by the authors:
Shandong Province National Natural Science Foundation: ZR2020MH149.
Scientific research project of Weifang Health Commission: WFWSJK-2022-231.

### Competing Interests

The authors declare there are no competing interests.

### Author Contributions

- Gaoxiu Qi conceived and designed the experiments, performed the experiments, analyzed the data, authored or reviewed drafts of the article, and approved the final draft.

- Jinmeng Liu analyzed the data, prepared figures and/or tables, contributed reagents, materials, analysis tools, and approved the final draft.
- Shuqi Tao performed the experiments, prepared figures and/or tables, and approved the final draft.
- Wenyuan Fan performed the experiments, prepared figures and/or tables, and approved the final draft.
- Haoning Zheng performed the experiments, prepared figures and/or tables, and approved the final draft.
- Meihong Wang analyzed the data, authored or reviewed drafts of the article, and approved the final draft.
- Hanchao Yang conceived and designed the experiments, authored or reviewed drafts of the article, and approved the final draft.
- Yongting Liu analyzed the data, prepared figures and/or tables, and approved the final draft.
- Huancai Liu conceived and designed the experiments, authored or reviewed drafts of the article, and approved the final draft.
- Fenghua Zhou conceived and designed the experiments, authored or reviewed drafts of the article, and approved the final draft.

## Human Ethics

The following information was supplied relating to ethical approvals (*i.e.*, approving body and any reference numbers):

This study was approved by the Ethics Committee of Affiliated Hospital of Weifang Medical University, China (approval number: wyfy-2022-ky-169).

## Data Availability

The raw measurements are available in the Supplementary Files.

## Supplemental Information

Supplemental information for this article can be found online at http://dx.doi.org/10.7717/peerj.15675#supplemental-information.

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
