# Peer review of "A retrospective study on expression and clinical significance of PHH3, Ki67 and P53 in bladder exophytic papillary urothelial neoplasms"

_PeerJ, doi:10.7717/peerj.15675_

## Round 0.1 · original submission · Major Revisions

All three reviewers gave their suggestions for modification. Please do your best to correct the manuscript and answer every question carefully.

Reviewer 1 ·

Basic reporting

The manuscript by and coworkers titled “Expression and clinical significance of PHH3, Ki67 and P53 in bladder exophytic papillary urothelial neoplasms” describes expression of phosphorylated Histone H3 (PHH3) in exophytic papillary urothelial neoplasms (EPUN). Interestingly, mitosis count based on PHH3 immunostaining (PHMC) is as good as hematoxylin and eosin staining based mitosis count (HEMC) in accuracy while it is significantly faster than HEMC. The pathologists seem to have performed thorough reading of PHH3 IHC slides, but language issues made the description of their observations confusing. Despite the issues with quality of the language, presentation and interpretation, this data is very interesting. This manuscript requires significant rewriting and rearrangement.

1. The major issue with this manuscript is language and presentation. This is leading to lack of clarity and misinterpretations. Improving the language would greatly help in describing the results clearly. It is beyond the scope of this review to enumerate all the corrections needed for this manuscript. For example, methods of immunohistochemistry on cancer tissues have been described in several previous papers. The paragraph (line 135-143) describing this, in fact was very poorly written despite authors having several papers available online for using as a template. EnVision methods? is it from Dako? Please mention the kit name and manufacturer. Wax blocks? Are these formalin fixed paraffin embedded blocks? Were the sections dehydrated or rehydrated? Antigen was repaired or retrieved? What is primary antibody dilution used? “Neutral gum was sealed”. Is it used for mounting the slides?
2. Even subheadings need rewriting. For example, lines 164-165.
3. Please explain how G2 cells were identified and why are you sure that those cells are not in any other cell cycle state. Please provide a rationale for excluding certain cells from your PHMC counts to prove that there is no bias in determination of PHMC.
4. The rationale for using PHH3 IHC staining for diagnosis and how it could be implemented is unclear. Although I greatly appreciate that there is significant difference in different subtypes of EPUN, its diagnostic efficacy determined in its current study design seems to be overplaying.
5. How were patients dichotomized according to recurrence in figure 6F. What is the minimum follow up period to categorize a patient as “no recurrence”?

Experimental design

No Comment.

Validity of the findings

No Comment.

Reviewer 2 ·

Basic reporting

In their manuscript by Gaoxiu Qi, Jinmeng Liu et al. investigate the suitability of various proliferation markers in diagnosing and determining the extent of exophytic papillary urothelial neoplasms. The manuscript is well-written in clear and unambiguous language.

Experimental design

The research question is well defined. The experimental design is sufficiently detailed described and the sample size adequate. The experiments were performed systematically.

Validity of the findings

The work described in this manuscript is of general interest.

The data provided supports the conclusion drawn in this manuscript.

Additional comments

1) Which type of PHH3 antibody was used for this study (Ser10 or Ser28). This is important as this effects the number positive cells counted in the sample. See Sun et al. “Level of phosphohistone H3 among various types of human cancers”
2) Furthermore only the company name and not the actual catalogue number for all the antibodies have been provided. This hinders others to apply the same assay on their sample.
3) Regarding Figure 4 was the same sample used to compare the different antibodies? For example are the images in 1A, 2A, 3A and 4A from the same HGPUC sample. This is not apparent from the manuscript.
4) In Figure 5 multiple p-value are associated with one scatter plot. It is not clear what these p-values are supposed to express.
5) Regarding line 197 to 204 it would be better to show this in a graph (e.g. positive stained cells per FOV).
6) In line 356 it is supposed to be time-saving, taking less time, instead of as it is stated in the text “time-consuming”.

Reviewer 3 ·

Basic reporting

See experimental design.

Experimental design

This study investigated expression of PHH3 in exophytic papillary urothelial neoplasms (EPUN) tissues. This is a retrospective study. Mitosis counts estimated in PHH3 IHC are equally efficient as counts measured in the gold standard H&E staining. The advantage with PHH3 IHC is the less time taken for pathologists for reporting mitosis counts. This is strength of the manuscript. The weakness of the study is diagnosis component. This part of the manuscript unclear and addressing this may require a meticulously designed future study. Pitching PHH3 as diagnostic marker in this manuscript may not be a good idea.
Please address the following comments.
a. All EPUN patients would not need complex diagnosis process for determining histological origin and subclassification. An ideal study design for addressing this issue would be to plot ROC curves for only those patients who were difficult to diagnose. The ROC curve data shown in the manuscript is on all patients. It is not clear how well PHH3 performs on patients with difficult diagnosis. However, the sample size in this study may not be adequate to perform this analysis. In such case, I ask authors to remove this part from the manuscript. If authors choose to show this data, please do the following. Explain the diagnosis problem clearly. Perform multinomial logistic regression and ROC curves. Show criterion value and regression formula for all categories. Show sensitivity, specificity, accuracy, PPV and NPV for all categories. Clearly identify, what are the features associated with misclassified patients in such analysis.
b. Please improve the language to accurately represent the science. Often, sentences are confusing and misrepresenting the data.
c. New daughter cell may not be produced in telophase. In telophase, nucleus divides into two and the daughter cells form after cytokinesis.
d. Please state if the correlations are positive in the results and in abstract.
e. Mention that it is a retrospective study in the abstract and mention sample size.
f. What is bottom of the epithelium? Do authors intend to say close to “the basement membrane”?
g. Write spearman rho value for correlations along with the p values in abstract.

Validity of the findings

See experimental design.

---

## Round 0.2 · Minor Revisions

Colleagues in the editorial department have raised questions about the data in Figure 7 and Table 3 and I have been unable to reproduce Figure 7 using the data in the Excel file.

Please **ALSO** provide the GraphPad Prism file you used to generate the survival curves.

Reviewer 1 ·

Basic reporting

My comments were addressed. I have no further comments.

Experimental design

--

Validity of the findings

--

Reviewer 2 ·

Basic reporting

No Comment

Experimental design

No Comment

Validity of the findings

No Comment

Reviewer 3 ·

Basic reporting

The authors addressed my comments satisfactorily.

Experimental design

...

Validity of the findings

...

---

## Round 0.3 · accepted · Accept

The author has corrected the incorrect data. I have found no other potential publication risks, so this manuscript can be published.